# Clinical, Epidemiological, and Geospatial Characteristics of Patients Infected with Hepatitis C Virus Treated with Second-Generation Direct-Action Antivirals in a Reference Center in a Mesoregion of São Paulo State, Brazil

**DOI:** 10.3390/microorganisms8101575

**Published:** 2020-10-13

**Authors:** Danilo Zangirolami Pena, Murilo Fernandes Anadão, Edilson Ferreira Flores, Mayara Namimatsu Okada, Alexandre Martins Portelinha Filho, Rodrigo Sala Ferro, Luiz Euribel Prestes-Carneiro

**Affiliations:** 1Infectious Diseases Outpatient Clinic, Oeste Paulista University, Presidente Prudente 19050-920, São Paulo, Brazil; danilo_zangirolami@hotmail.com (D.Z.P.); murilofanadao@outlook.com (M.F.A.); amportelinha@gmail.com (A.M.P.F.); rodrigosalaferro@hotmail.com (R.S.F.); 2Statistics Department, School of Sciences and Technology, São Paulo State University, Presidente Prudente 19060-900, São Paulo, Brazil; edilson-ferreira.flores@unesp.br; 3Pharmacy of the Outpatient Medical Specialties of Presidente Prudente, Presidente Prudente 19050-680, São Paulo, Brazil; mayara.fmepp@gmail.com

**Keywords:** sustained virological response, comorbidities, gastrointestinal, depression, incidence

## Abstract

Hepatitis virus infection is a major public health problem worldwide. Currently, Brazil has almost 700,000 cases. The Brazilian Unified Health System (SUS) provides therapeutic regimens for people infected with hepatitis C virus (HCV). We determined the clinical, laboratory, epidemiologic, and geospatial characteristics of patients infected with HCV treated with second-generation direct-action antivirals (DAAs) in a hospital reference center in São Paulo state, Brazil, using data from file records. A map was constructed using a geographic information system. From 2015 to 2018, 197 individuals received second-generation DAAs (mean age, 57.68 ± 1.36 years; interquartile range, 56.22–59.14 years; 58.9% male; 41.1% female). Genotypes 1a and 1b accounted for 75.7% of cases and the prevalent therapeutic regimen was sofosbuvir/simeprevir. Sustained viral response accounted for 98.9% and the METAVIR score F3/F4 for 50.8%. Increased alanine transferase was significantly correlated with an increase in α-fetoproteins (*p* = 0.01), and severe necro-inflammatory activity (*p* = 0.001). Associated comorbidities were found in 71.6%, mainly coronary artery and gastrointestinal disorders. The cumulative incidence in the region was 2.6 per 10,000 inhabitants. Our data highlight the role of reference hospitals in Brazil’s public health system in the treatment of HCV. Low incidence rates demonstrated the fragility of municipalities in the active search for patients.

## 1. Introduction

Viral hepatitis infection is a major public health problem worldwide. It is responsible for an estimated 1.4 million deaths per year from acute infection and hepatitis-related liver cancer and cirrhosis. Of those deaths, approximately 47% are attributable to hepatitis B virus (HBV) and 48% to hepatitis C virus (HCV) [1]. In Brazil, the prevalence of HCV antibodies (anti-HCV) is close to 700,000 [2]; however, it is possible that this number is only a quarter of the total population actually infected [3].

Brazil is one of the exponents of HCV treatment. The Brazilian Ministry of Health provides the most recent drugs available based on the best cost effectiveness. There have been dramatic changes since the use of polyethylene glycol-interferon and ribavirin, including high rates of non-sustained viral response (NSVR) and intense side effects [4]. The first generation of direct-action antivirals (DAAs), boceprevir/telaprevir, both used in combination with alfa-peginterferon plus ribavirin-PEG-IFN + RBV (PR), thus constituting a triple therapy (PR + IP) increased the sustained viral response (SVR) but were associated with intolerable side effects [5]. Second-generation DAAs (sofosbuvir (SOF)/daclatasvir (DCV)/simeprevir (SIM)), available since 2014 in the Brazilian Unified Health System (SUS), have been developed to ensure better results. With the approval and release of second-generation DAAs free from interferon by the SUS, the use of interferon-based regimens was gradually replaced [6]. Such a strategy is fundamental to the success of the Plan for the Elimination of Hepatitis C in Brazil as a public health problem by 2030 [7]. These goals are in line with the strategy adopted by the World Health Organization, which ultimately establishes global strategies, including reduction of the incidence of HCV by 80% and HCV-related mortality by 65% by 2030 [8].

As a complication of chronic hepatopathy, it has been shown that patients infected with HCV have a higher prevalence of comorbidity and multimorbidity [9]. Although viral replication in different organs has been demonstrated, the underlying mechanisms are not completely understood [10,11].

In epidemiologic and public health studies, geospatial analysis has been used successfully for vector-borne diseases regarding environmental health, risk assessment, and injury prediction, and for the development of health policies and intervention planning [12,13]. In Brazil, clinical and epidemiologic studies on individuals chronically infected with HBV and HCV showed that capitals and large cities are the focus of infection in specific vulnerable groups. Geospatial tools show that the distribution of cases within the general population living in the countryside is low. We determined the clinical, epidemiologic, and geospatial characteristics of patients infected with HCV treated with second-generation DAAs in a reference center in a mesoregion in the countryside of São Paulo state, Brazil.

## 2. Materials and Methods

### 2.1. Study Design and Setting

In 2019, São Paulo, the richest and most populous state of Brazil, with an estimated population of 45,919,049 inhabitants, accounts for 21.8% of the entire population of Brazil, estimated to be 210,147,125 inhabitants according to the Brazilian Institute of Geography and Statistics (IBGE). Geographically, the state is divided into 15 mesoregions and 18 regional health care networks (RHCNs) for administrative health care. The western region of São Paulo state, with 45 municipalities and an estimated population of 753,344 in 2018 (IBGE), is considered to be one of the poorest and asymmetric regions of São Paulo state [13]. The region is characterized by small municipalities with <20,000 inhabitants. The western region comprises Regional Health Assistance Network11 (RHAN11), located in Presidente Prudente mesoregion 8. The city is a mid-sized urban center 560 km from the state capital, São Paulo. In 2019, the estimated population was 228,743 inhabitants (IBGE) [13]. The Regional Hospital of Presidente Prudente (RH), located in Presidente Prudente, is a 550-bed tertiary-care public university hospital providing care for SUS patients, and since 2000, it has been a reference center for the diagnosis and treatment of viral hepatitis caused by HBV and HCV [14]. From January 2000 to December 2013, a percentage of patients diagnosed with chronic HCV in municipalities of RHAN11 were referred to the infectious disease’s outpatient clinics of RH and others were diagnosed and treated in settings such as private clinics and public health care centers. However, the therapeutic regimen was dispensed freely by the pharmacy at RH, linked to SUS. Due to increasing numbers of patients, a specific outpatient clinic for viral hepatitis was created in January 2014 in RH, and the dispensing of the therapeutic regimen was transferred to the pharmacy for outpatient medical specialties (Ambulatório de Especialidades Médicas (AME)) for Presidente Prudente, maintained by SUS. Patients with HCV antibodies (anti-HCV) were referred to the outpatient clinic for viral hepatitis at RH by primary health care centers in RHAN11 municipalities. This is a transverse and retrospective study conducted from January 2015 to December 2018 on patients treated in RH with second-generation DAAs.

### 2.2. Inclusion/Exclusion Criteria

We reviewed the medical records of patients >18 years old with confirmed HCV infection (anti-HCV+ and HCV-RNA+) according to the Clinical Protocol and Therapeutic Guidelines for the Treatment of Chronic Hepatitis C and Co-infections [6], or previous guidelines from Brazil’s Ministry of Health, and who were prescribed second-generation DAAs. We excluded patients with confirmed HCV infection who discontinued treatment or received their treatment at another institution, or whose data were incomplete.

### 2.3. Data Collection and Baseline Laboratory Parameters

The baseline information was collected from file records before treatment with second-generation DAAs and consisted of demographic variables (age and sex), genotype, liver fibrosis (METAVIR score), HCV viral load before and 1 year after treatment, prescribed regimen, treatment duration (12 or 24 weeks), coinfections (human immunodeficiency virus (HIV) or HBV or syphilis). Details of associated comorbidities were obtained from self-reports by the patients and from laboratory biopsy and imaging results, as well as from the different specialties, including psychiatry, gastroenterology, hematology, clinical surgery, and cardiology. Psychiatric disorders and therapeutic regimens were self-reported by the patients.

Hepatitis serology was performed using the enzyme-linked immunosorbent assay for HCV, HBV, and HIV. Syphilis infection was determined by the Venereal Disease Research Laboratory test and fluorescent treponemal antibody-absorption test. HCV viral load and genotype were determined according to the manufacturer’s instructions in the Adolfo Lutz Institute, the former public health laboratory of São Paulo state. Hemograms were performed using a flow cytometer flux counter. Alanine aminotransferase (ALT), aspartate aminotransferase (AST), albumin, globulins, bilirubin, alkaline phosphatase, γ-glutamyl transferase (γ-GT), α-fetoprotein (AFP), urea, and creatinine were assessed using automated systems according to the manufacturer’s instructions.

The prescribed regimen and treatment duration were established by the Clinical Protocol and Therapeutic Guidelines for the Treatment of Chronic Hepatitis C and Co-infections [6]. Although second-generation DAAs were available since 2014 in SUS, in Presidente Prudente, the use of interferon-free regimens did not start until 2015. Liver fibrosis was determined by anatomopathological analysis of a liver fragment removed by core biopsy, or by elastography, and the METAVIR score was assessed. To assess whether or not SVR was reached, the patient had to have detectable levels of a viral load pretreatment, take the prescribed treatment with appropriate drugs for the required time, and have an undetectable viral load 1 year after the end of therapy. Patients who received incorrect doses or drugs, did not take the proposed treatment for the required time, or did not undergo clinical follow-up before and after treatment were not included in the analysis to affirm the efficacy of the drugs.

### 2.4. Geospatial Analysis and Construction of Maps

For geospatial analysis, maps were constructed using geographic information system (GIS) technology and spatial statistics. In spatial statistics, the study of punctual processes was explored, focusing on the spatial distribution of the residence of patients by municipality in RHAN11. In the GIS technology, ArcGIS module visualization was used as developed by the Environmental Systems Research Institute (ArcGIS software 10.2.2; ESRI, Redlands, CA, USA). Both processes transcribe the information from the database to the maps, giving a broad view of the phenomena and their concentrations.

### 2.5. Statistical Analysis

The results are shown as means ± standard error of the mean (for normally distributed variables) with 95% confidence intervals. Dichotomous and nominal variables are expressed as frequencies and percentages. The patients’ municipality of residence was obtained from file records to investigate the geospatial distribution in the RHAN11 mesoregion. The cumulative incidence rates were the total number of patients treated for HCV in each municipality. For the period from 2015 to 2018, the crude incidence coefficient was calculated by dividing the number of reported cases by the resident population in the same place and time, and multiplying the result by 10,000. The cumulative incidence rate for RHAN11 was the total number of patients treated with HCV divided by the entire population, per 10,000 inhabitants, during the same period. Statistical analysis was performed using GraphPad Software (San Diego, CA, USA) and the Sigma-Stat program (Systat Software, Richmond, CA, USA). Multiple linear regression analysis was used with four predictor variables (*X*) and a continuous response (*Y*) using the least-squares estimate and Minitab software from the Laboratory of Applied Statistics of the Statistics Department of UNESP/FCT, São Paulo, Brazil. Maps were constructed using GIS.

### 2.6. Ethical Statement

All procedures performed in studies involving participating human subjects were done in accordance with the ethical standards of the institution and/or national research committee and with the 1964 Helsinki Conference declaration and its subsequent amendments or comparable ethical standards. The study was in accordance with the ethical standards of the Institutional Ethics Committee of Oeste Paulista University, Presidente Prudente, São Paulo, Brazil (ID, 58691016.2.0000.5515).

## 3. Results

### 3.1. Therapeutic Regimens, Baseline Clinical, and Demographic Characteristics

Treatment for patients chronically infected with HCV in the Reference Center of the Regional Hospital of Presidente Prudente was introduced in 2010, and 751 patients were investigated between 2010 and 2018. Figure 1 shows the total number of patients who underwent HCV treatment, an interferon-based regimen (*n* = 368, 49.0%), and an interferon-free regimen (*n* = 383, 51.0%). Figure 1 also shows that 383 patients received second-generation DAAs between 2015 and 2018, and 146 (48.6%) were treated in other settings, including private clinics, for which no epidemiological or therapeutic data are available. One hundred and ninety-seven (51.4%) individuals were treated at the pharmacy for outpatient medical specialties (AME). All of these patients finished the proposed treatment and were followed for determination of viral load 1 year after the end of the treatment. Figure 2 shows the evolution of the therapeutic regimens for the treatment of HCV. The therapeutic regimens followed the guidelines of Brazil Health Ministry. Figure 3 shows the number of patients for which therapeutic regimens were dispensed in Presidente Prudente annually. From 2010 to 2014, it was not possible to differentiate if the patient was treated at RH, at a private clinic, or other public health care center, or which therapeutic regimen was adopted. From 2015 to 2018, it was possible to distinguish those treated at RH and those treated in other services. The highest number of patients were treated in 2016 and the levels decreased in 2017 and 2018. Between January 2015 and December 2018, of 383 patients investigated, 197 received second-generation DAAs (Figure 1 and Figure 2). The mean age of the whole group was 57.68 ± 1.36 years (interquartile range, 56.22–59.14 years; varying from 32 to 86 years), distributed as follows: 31–40 years, 9 (4.6%); 41–50 years, 37 (18.8%); 51–60 years, 67 (34.0%); 61–70 years, 62 (31.5%); 71–80 years, 19 (9.6%), and 81–90 years, 3 (1.5%). Regarding gender, 116 (58.9%) were male and 81 (41.1%) were female, a ratio of 1.4:1.

Genotypes 1a and 1b accounted for 75.7% of the cohort and SOF/SIM was the most therapeutic regimen, followed by SOF/DCV; treatment duration was 12 weeks for most patients. METAVIR score F3/F4 accounted for 50.8% of the population, although liver biopsy or elastography was not performed on most patients. Sixty-five patients (33%) failed previous treatments, some of which involved more than one regimen to control the HCV infection, and they were retreated with second-generation DAAs. After treatment with second-generation DAAs, SVR was achieved in 98.9% of patients (Table 1). Coinfections with other sexually transmitted infections were found in 11 of 197 patients (5.6%): HCV/HIV in 7 of 197 (3.5%) and HCV/syphilis in 4 of 197 (2%).

### 3.2. Laboratory Findings

Laboratory data were determined before treatment with second-generation of DAAs and few parameters increased in the cohort. Among the most important laboratory data, increased levels of ALT, AST, γ-GT, and AFP were found. AFP was increased in 48 of 197 patients (24.3%) (Table 2). In a multiple linear regression analysis, increased ALT was significantly correlated with increased levels of AFP (*p* = 0.01) and severe necro-inflammatory activity (METAVIR score F3/F4; *p* = 0.001). No correlation was found between ALT and other increased variables (γ-GT and AST).

### 3.3. Comorbidities Associated with Chronic HCV Infection

Comorbidities were found in 141 of 197 (71.6%) of patients infected with HCV. Table 3 shows the most prevalent diseases associated with chronic HCV infection, determined before treatment with second-generation DAAs. Most patients had more than one associated comorbidity. Coronary artery disorders were the most common extrahepatic disease and were found in 76 of 197 (38.6%) of the patients; hypertension was found in 65 of 197 patients (33%), heart failure in 7 of 197 (3.6%), acute coronary syndrome in 3 of 197 (1.5%), and tachycardia in 1 of 197 (0.50%). Gastrointestinal disorders were the second most common extrahepatic disease and were found in 74 of 197 patients (37.5%); esophageal varices were found in 22 of 197 (11.2%), splenomegaly in 14 of 197 (7.1%), cholelithiasis in 13 of 197 (6.6%), and ascites in 10 of 197 (5.0%). Hormonal and metabolic disorders were the third most prevalent extrahepatic disease; diabetes was found in 40 of 197 (20.3%) patients. Psychiatric disorders were the fourth most frequent disease, found in 39 of 197 patients (19.8%) of which depression was present in 23 of 197 patients (11.7%), followed by schizophrenia in 7 of 197 (3.5%), anxiety in 3 of 197 (1.5%), suicide attempt and bipolar disorder in 2 of 197 (1.0%), and alcohol abuse and insomnia in 1 of 197 (0.5%). Among the 39 patients with psychiatric disorders, antidepressants and selective serotonin reuptake inhibitors were prescribed for 21 of 39 (53.8%), escitalopram/citalopram for 6 of 39 (15.4%), amitriptyline for 5 of 39 (12.8%), sertraline for 5 of 39 (12.8%), fluoxetine for 3 of 39 (7.7%), paroxetine for 1 of 39 (2.6%), and clomipramine for 1 of 39 (2.6%). Benzodiazepines were prescribed for 15 of 39 (38.5%), clonazepam for 10 of 39 (25.6%), lorazepam for 2 of 39 (5.1%), diazepam for 1 of 39 (2.6%), bromazepam for 1 of 39 (2.6%), and alprazolam for 1 of 39 (2.6%). Neuroleptics were prescribed for 11 of 39 (28.2%), risperidone for 5 of 39 (12.8%), levomepromazine for 3 of 39 (7.7%), haloperidol for 2 of 39 (5.1%), and chlorpromazine for 1 of 39 (2.6%). The mood stabilizer, carbamazepine, was prescribed for 1 of 39 (2.6%). Ophthalmic diseases were the fifth most prevalent comorbidity and were found in 32 of 197 patients (16.2%) and included presbyopia in 7 of 197 (3.6%), astigmatism in 7 of 197 (3.6%), followed by cataract in 5 of 197 (2.5%). Thyroid diseases were the sixth most prevalent comorbidity and were found in 19 of 197 patients (9.6%) with hypothyroidism present in 15 of 197 patients (7.6%). Other comorbidities are shown in Table 3. Risk behaviors include risk factors of nonadherence, disappointing treatment outcomes, or unsubstantiated beliefs about reinfection. Furthermore, their association with HCV aggravates the inflammation, worsening the liver disease and increasing the prevalence of hepatocellular carcinoma (HCC). Among the main risk factors related by the patients, smoking was reported in 32 of 197 (16.2%); alcohol abuse in 19 of 197 (9.6%); mental disorders including schizophrenia in 7 of 197 (3.5%); and history of drug injection or illicit drug use was found in 6 of 197 (3.0%).

### 3.4. Geographic Distribution of Genotypes and Cumulative Incidence of HCV in the RHAN11 Mesoregion

In Figure 4, populations > 40,000 inhabitants in 2018 are marked with an asterisk. The municipalities were identified by number in alphabetical order. In 37 of the 45 (82.2%) municipalities in RHAN11, patients infected with HCV were found from 2015 to 2018. Two municipalities showed higher accumulated incidence, with 11–16 infected individuals per 10,000 inhabitants. Three municipalities showed an incidence between 6 and 10; most municipalities showed an incidence between 1 and 5 infected individuals per 10.000 inhabitants. Genotype 1a had a broad distribution; however, in the city of Rosana in the south, with a low number of inhabitants, all genotypes were identified. Based on the overall population living in the 45 municipalities of RHAN11, the cumulative incidence was 2.6 per 10,000 inhabitants.

## 4. Discussion

Cohort community-based studies may represent a more significant premise for evaluating disease dynamics in a population. In western São Paulo state, a poor and asymmetric region and considered the last frontier of development, from 2015 to 2018, 383 patients with chronic HCV infection were treated with second-generation DAAs. One hundred ninety-seven patients were treated in RH, and 98.9% reached SVR. A geospatial analysis showed that 82.2% of the 45 municipalities in the region harbored patients infected with HCV. The results highlight the role of reference public tertiary hospitals in the universal treatment of HCV infection in the public health system of Brazil.

From 2010 to 2018, 751 patients with chronic HCV received a therapeutic regimen in the RH/AME complex; from 2015 to 2018, genotypes 1a and 1b were the most prevalent among the 197 patients investigated, a result previously found in patients infected with HCV who were followed in the RH from 2000 to 2006 [15]. Similar results were found in a survey representative of the whole country, which showed that genotype 1a was more prevalent in the south, southeast, and central regions, whereas genotype 1b was more prevalent in the northern regions [16]. Genotype 1 was also reported to be the most prevalent in different regions of Europe and in Central Asia [17,18]. Treatment of HCV can be complex because the genotypes do not respond in the same way to some therapies, and the class of genotype defines the therapeutic regimen. SOF/SIM and SOF/DCV accounted for most of the treatments in our cohort, with duration of 12 weeks in 77.7% of patients, and an SVR of 98.9% was achieved. Most of our patients (67%) were treatment naive and with METAVIR scores F0, F1, and F2. These characteristics certainly contributed to the higher levels of SVR found. Real-life clinical studies demonstrate that patients infected with HCV treated with second-generation DAAs achieve SVR rates above 90% [19]. In a highly representative population of patients infected with HCV throughout the country and treated with the same therapeutic regimen, the overall SVR rates were higher than 95% [20]. In a real-world cohort from three hepatology centers from southern Brazil, SVR was achieved in 95% of patients treated with second-generation DAAs [21]. In Spain, rates of 96.8% and 95.8% were found in two cohorts of individuals infected with HCV with genotype 1 treated by second-generation DAAs [22]. Although different studies have their own characteristics and bias should be considered, we can conclude that one of the main advantages of the second-generation DAA therapeutic regimen is the high rate of SVR obtained worldwide.

Laboratory data are a key factor in the diagnosis and follow-up of patients infected with HCV. In our cohort, at baseline levels, few parameters were outside the normal ranges. Increased levels of AST, ALT, γ-GT, and AFP were found. These results may be anchored by the scientific literature in which increased levels of serum aminotransferases (ALT/AST), overwhelmed by ALT levels, in untreated individuals with chronic viral hepatitis infection may be a predictor of progression of liver fibrosis. In these patients, when parenchymal liver cells are damaged, aminotransferases leak from the liver into the blood, resulting in increased levels of these enzymes in the bloodstream [23]. However, around 25% of patients with chronic HCV have persistently normal aminotransferase levels with mild or no symptoms [24]. In our cohort, 48 of 197 patients (24.3%) showed increased levels of AFP. AFP plays a significant role in predicting the recurrence and metastasis of HCC [25], but the role of AFP as a surveillance and diagnostic tool for HCC is controversial [26]. However, in resource-poor settings such as RH, AFP is one of the tools available for following up patients infected with HCV for progression to HCC. A significant correlation between increased levels of ALT, AFP, and necro-inflammatory levels (F3/F4) was found in our cohort. Recently, it was shown that the combination of increased ALT, AFP, presence of cirrhosis, and higher MELD score (model for end-stage liver disease) are determinants for the time to development of HCC. Taken together, they may constitute a tool for clinicians to screen for HCC before and after patients reach SVR [27].

One of the most important findings in our cohort, before treatment with second-generation DAAs, was the high number of comorbidities with rates of 71.6%. In clinical practice, clinicians are faced a great number of non-liver diseases associated with patients with chronic HCV, most of which are more serious than the hepatic disease. Among these patients, coronary artery disorders were the most prevalent, including hypertension, heart failure, acute coronary syndrome, and tachycardia. In chronic liver disease induced by different agents, the end stage of chronic HCV infection is cirrhosis. Portal hypertension is one of the consequences of liver cirrhosis that triggers the appearance of a series of complications such as esophageal varices, ascites, and splenomegaly [28]. Psychiatric disorders were the fourth most frequent comorbidity. Rates of 19.8% were found in patients infected with HCV compared with 1% in the general population, mainly depression and anxiety [29]. In Brazil, among patients infected with HCV in a university hospital in the northeastern region, 49% had at least one psychiatric diagnosis [30]. In a review on neurologic and psychiatric disorders associated with chronic HCV infection, psychiatric disorders were found in up to 50% of cases [31]. There is emerging evidence of HCV neuroinvasion. The virus has recently been detected in brain tissue at autopsy, raising the possibility that HCV infection of the brain could be directly related to the reported neuropsychological and cognitive changes [10]. However, other mechanisms proposed to explain the pathogenesis of neuropsychiatric disorders include derangement of metabolic pathways of infected cells, alterations in neurotransmitter circuits, autoimmune disorders, and cerebral or systemic inflammation [32]. As far as we know, this is the first study reporting on therapeutic regimens for psychiatric disorders in patients infected with HCV in Brazil.

Of the 45 municipalities in RHAN11, patients infected with HCV were found in 82.2%, highlighting that untreated patients infected with HCV may be spreading the virus in their communities. Most municipalities showed low levels of accumulated incidence, varying from 1 to 5 individuals per 10,000 inhabitants, and the cumulative incidence of the region was 0.26 per 100,000 inhabitants. This rate is about 10-fold lower than the 6.7 cases per 100,000 inhabitants found in Brazil between 2001 and 2012, and 45-fold lower than the incidence rates found countrywide in 2017 (11.9 cases per 100,000 inhabitants) [33]. The incidence rates found in Brazil are much lower than in developed countries such as Canada and the United States and countries in Europe, which have much more developed and efficient screening and notification systems [34,35]. The reasons why the incidence levels were lower countrywide and worldwide than in the western region are not well established. The Brazilian Health Ministry launched a countrywide campaign aimed at simplifying the diagnosis and expanding testing, especially in vulnerable populations through the HCV-rapid test, stimulating active searching for diagnosis in individuals not linked to SUS. However, it seems that these measures do not reach the public health system of the small and poorer cities of our region. In this context, the region also harbors other endemic vector-borne diseases linked to poverty and undeveloped settings, including dengue fever and visceral leishmaniasis [36].

The main shortcomings of our research are the difficulties associated with low levels of elastography in patients not indicated for liver biopsy, the lack of follow-up of patients after treatment due to long distances between the RH and municipalities, and difficulties in psychiatric follow-up in patients with psychiatric disorders. 

Our data have global relevance and may be compared with undeveloped regions of developing countries that face similar limitations, with the goal of a reference center for treating patients infected with chronic HCV, but have low incidence rates and difficulties with active searching for new patients. Furthermore, a source of bias in our study may be the absence of data for patients who received any dose of second-generation DAAs and discontinued treatment for any reason, such as death by HCC or other comorbidities, severe side effects, or did not attend the outpatient clinic after the end of the treatment.

## 5. Conclusions

In the western region of São Paulo, treatment of patients with chronic HCV infection with second-generation DAAs and the high levels of SVR obtained highlight the role of reference hospitals in the public health system of Brazil. The low incidence rates of HCV infection found in the region demonstrate the fragility of basic health units in the active search for patients to meet the target of HCV eradication by 2030, proposed by the Brazilian Ministry of Health.

## Figures and Tables

**Figure 1 microorganisms-08-01575-f001:**
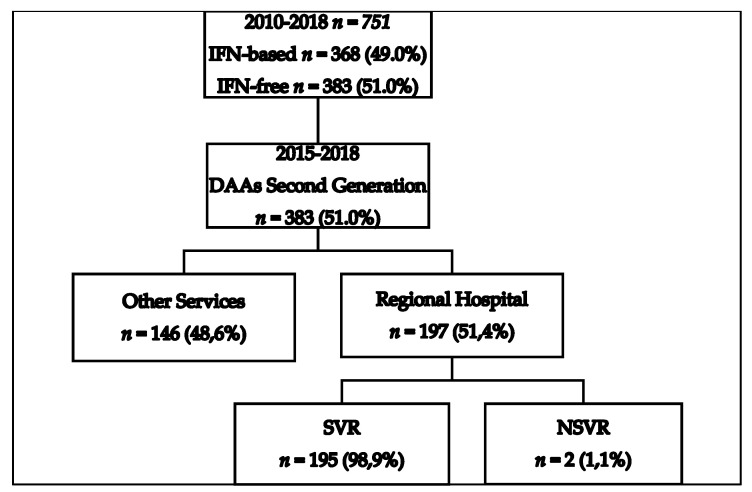
Flowchart of patients diagnosed with hepatitis C virus (HCV) in the Regional Health Assistance Network11 (RHAN11) mesoregion from 2010 to 2018 and treated with first- and second-generation direct-action antivirals (DAAs). IFN-based, interferon-based; NSVR, non-sustained viral response; SVR, sustained viral response.

**Figure 2 microorganisms-08-01575-f002:**
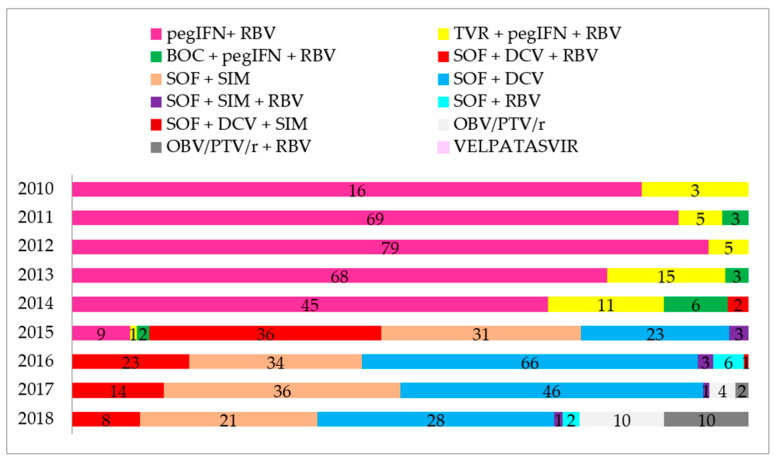
Evolution of HCV therapeutic regimens in a reference center in RHAN11 mesoregion located in Presidente Prudente, São Paulo, Brazil. BOC, boceprevir; DCV, daclatasvir; pegIFN, polyethylene glycol-interferon; RBV, ribavirin; SIM, simeprevir; SOF, sofosbuvir; TVR, telaprevir; OBV, ombitasvir; PTV, paritaprevir; r, ritonavir.

**Figure 3 microorganisms-08-01575-f003:**
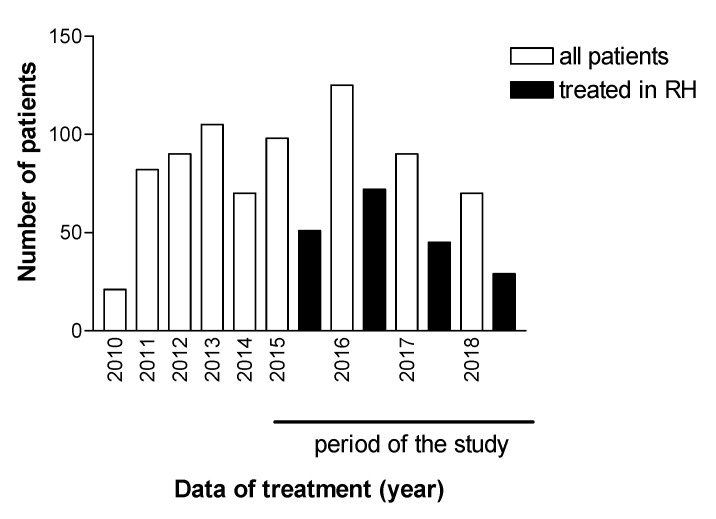
Patients diagnosed with HCV from the RHAN11 mesoregion and patients whose treatment was dispensed by the pharmacy of the Specialist Medical Clinic of Presidente Prudente and treated in the Regional Hospital (RH) of Presidente Prudente, São Paulo, Brazil.

**Figure 4 microorganisms-08-01575-f004:**
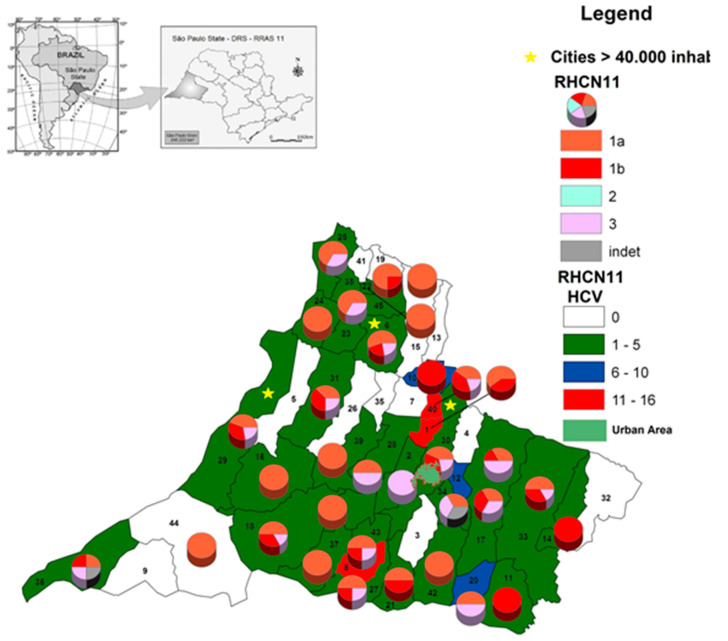
Geographic distribution of HCV and genotypes of patients diagnosed and treated between 2015 and 2018, according to the municipality of residence in RHAN11 mesoregion, western São Paulo, Brazil. The colors represent the cumulative number of cases. The municipalities are numbered in alphabetic order: 1. Alfredo Marcondes; 2. Alvares Machado; 3. Anhumas; 4. Caiabu; 5. Caiuá; 6. Dracena; 7. Emilianópolis; 8. Estrela do Norte; 9. Euclides da Cunha Paulista; 10. Flora Rica; 11. Iepê; 12. Indiana; 13. Irapuru; 14. João Ramalho; 15. Junqueirópolis; 16. Marabá Paulista; 17. Martinópolis; 18. Mirante do Paranapanema; 19. Monte Castelo; 20. Nantes; 21. Narandiba; 22. Nova Guataporanga; 23. Ouro Verde; 24. Panorama; 25. Paulicéia; 26. Piquerobí; 27. Pirapozinho; 28. Presidente Bernardes; 29. Presidente Epitácio; 30. Presidente Prudente; 31. Presidente Venceslau; 32. Quatá; 33. Rancharia; 34. Regente Feijó; 35. Ribeirão dos Índios; 36. Rosana; 37. Sandovalina; 38. Santa Mercedes; 39. Santo Anastácio; 40. Santo Expedito; 41. São João do Pau d’Alho; 42. Taciba; 43. Tarabai; 44. Teodoro Sampaio; 45. Tupi Paulista.

**Table 1 microorganisms-08-01575-t001:** Baseline clinical and demographic characteristics of patients with chronic hepatitis C (*n* = 197).

Characteristic	Number (%)
Genotype	
1a	93 (47.2)
1b	56 (28.5)
2	2 (1.0)
3	44 (22.3)
Indeterminate	2 (1.0)
Therapeutic regimen	
SOF/DCV	71 (36.0)
SOF/DCV/RBV	40 (20.3)
SOF/RBV	3 (1.5)
SOF/SIM	78 (39.6)
SOF/SIM/RBV	5 (2.5)
Treatment duration	
12 weeks	153 (77.7)
24 weeks	44 (22.3)
Liver fibrosis	
F0	5 (2.5)
F1	40 (20.3)
F2	31 (15.7)
F3	21 (10.6)
F4	47 (23.9)
Unknown	53 (26.9)
Previous treatment	
Previous failed treatments	65 (33)
Naive	132 (67)
Viral response	
SVR	195 (98.9)
NSVR	2 (1.0)
Average viral load (IU/mL)	1,218,994

SOF, sofosbuvir; DCV, daclatasvir; RVB, ribavirin; SIM, simeprevir; SVR, sustained virologic response; NSVR, non-sustained virologic response.

**Table 2 microorganisms-08-01575-t002:** Baseline laboratory parameters of the patients with chronic hepatitis C.

Parameters	Number	Mean ± SD	95% CI	Normal Range
Hemoglobin (g/dL), women	80	13.26 ± 1.75	12.87–13.65	12.0–15.5
Hemoglobin (g/dL), men	113	14.64 ± 0.33	13.98–15.31	13.5–17.5
Hematocrit (%) women	80	40.04 ± 0.51	39.00–41.07	36–48
Hematocrit (%) men	113	42.26 ± 0.54	41.18–43.35	41–50
Platelets < 150 (mm^3^)	192	157.60 ± 76.21	146.80–168.50	150–400
TAP (seconds)	171	12.23 ± 4.95	11.48–12.98	10–14
INR	174	1.15 ± 0.76	1.03–1.26	0.8–1.0
Albumin (mg/dL)	177	4.121 ± 2.45	3.75–4.48	3.4–5.4
Globulins (mg/dL)	126	3.60 ± 0.61	3.49–3.71	2.3–3.5
Total bilirubin (mg/dL)	189	0.97 ± 0.86	0.85–1.10	0.1–1.2
ALT (IU/dL)	194	91.73 ± 67.04	82.24–101.2	7–56
AST (IU/dL)	194	78.47 ± 55.22	70.65–86.29	10–40
Alkaline phosphatase (IU/L)	108	107.60 ± 62.62	95.68–119.6	44–147
γ-GT (IU/L)	159	130.2 ± 136.3	108.8–151.8	85
α-Fetoproteín (ng/mL)	167	12.78 ± 28.48	8.42–17.13	<10
Urea (mg/dL)	157	29.09 ± 15.64	26.62–31.56	16–40
Creatinine (mg/dL)	190	0.85 ± 0.60	0.76–0.93	0.6–1.2

SD, standard deviation; CI, confidence interval; TP, prothrombin time; INR, international normalized ratio; ALT, alanine aminotransferase; AST, aspartate aminotransferase; γ-GT, γ-glutamyl transferase.

**Table 3 microorganisms-08-01575-t003:** Comorbidities associated with chronic HCV infection in patients treated in a reference center between 2015 and 2018.

Comorbidities	Number (*n* = 197)	%
Coronary artery disorders	76	38.6
Gastrointestinal	74	37.5
Hormonal and metabolic disorders	40	20.3
Psychiatric disorders	39	19.8
Ophthalmic disorders	32	16.2
Thyroid disorders	19	9.7
Kidney disorders	13	6.6
Neurologic disorders	8	4.0
Hematologic disorders	8	4.0
Autoimmune disorders	8	4.0
Tumors disorders	6	3.0
Lung disorders	2	1.0
Skin allergy	2	1.0

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
