# Peer review of "Clinical, Epidemiological, and Geospatial Characteristics of Patients Infected with Hepatitis C Virus Treated with Second-Generation Direct-Action Antivirals in a Reference Center in a Mesoregion of São Paulo State, Brazil"

_microorganisms, 2020, doi:10.3390/microorganisms8101575_

Round 1
Reviewer 1 Report
- In lines 47-51, “The first generation of direct-action antivirals (DAAs) (boceprevir/telaprevir) increased the sustained viral response (SVR) but were associated with intolerable side effects [5]. Second-generation DAAs (sofosbuvir [SOF]/daclatasvir [DCV]/simeprevir [SIM]), available since 2014 in the Brazilian Unified Health System (SUS), have been developed to ensure better results [6].” Authors should describe these treatments in detail here. For an example, make corrections from “telaprevir” to “telaprevir combination with peginterferon plus ribavirin.” When did you start to use interferon-free regimens?
- Authors should describe your patients clearly. In Figure 1, authors should mention about patients with interferon-free treatment and those with interferon-including treatment.
- Is there any patients who were re-treated?
- In Figure 2, authors should describe the number of patients.
Author Response
Attached file with the amendments recommended by the reviewer 1

Reviewer 2 Report
The availability of highly effective oral HCV treatment with few side effects, known as DAAs, makes HCV cure possible in nearly all infected patients. This study shows, the efficacy of treating HCV, as a strategy to the success of the Plan for the Elimination of Hepatitis C in Brazil. However, this study has three major concerns:
- Nonhepatic comorbidities, such as kidney diseases, diabetes, and coronary artery disease are common in patients with HCV, across all levels of hepatic fibrosis. All of them are likely to affect clinical course and may complicate HCV care management. None of these comorbidities associated with chronic HCV infection is showed in table 3. There is any explanation?
- Data on risk behaviours is not provided – unsafe injections, injection drug use, abused alcohol, and mental health disorders are risk factors of non adherence, and disappointing treatment outcomes, or unsubstantiated beliefs about reinfection. There is no data for hepatitis C risk behaviours in this study.
- All patients that received any dose of treatment, and discontinued treatment, for any reason, were excluded. So, there is a bias on the results of the sustained viral response achieved.
Author Response
Attached file with the amendments recommended by the reviewer 2

Round 2
Reviewer 1 Report
All queries were fixed.
Author Response
Reviewer 1
Dear reviewer, thanks for your kindness in analyzing the corrections made in our manuscript and your approval. Kind regards, Prof. Luiz Euribel Prestes Carneiro MD, PhD
Reviewer 2 Report
Lines 29-30 – Taking in account the change in table 3, the percentage of nonhepatic comorbidities, and the order of the most common of them may change. The new comorbidities order is presented in the table 3.
Line 193 (figure 2) – Instead SOF+DVC may be SOF+DCV.
Lines 197-210 – The data added to the paper, complementing the information of figure 2 does not add value to the manuscript it is suggested to remove.
Line 248 – Taking in account the changed comorbidities, did the number and percentage of patients with nonhepatic comorbidities, also, were not changed?
Line 251 – The number of coronary artery disorders in the text do not matched with the number presented in the table 3.
Line 317 – …from 2000 to 2006 [16]. Similar results…
Line 320 – …regions of Europe and in Central Asia [17,18].
Line 276 – …and increasing the prevalence of hepatocellular carcinoma (HCC).
Lines 346-347 – …and metastasis of HCC [25],…
Line 356 – In the abstract the comorbidities rate is 64.4%, and mentioned as 64.5%. Please clarify.
Lines 358-359 – Gastrointestinal disorders were not the most prevalent nonhepatic comorbidities. Please correct
Lines 362-363 – Psychiatric disorders were not the second most prevalent nonhepatic comorbidities. Please correct.
Author Response
Reviewer 2
Comments and Suggestions for Authors
Dear reviewer, thanks again for your kindness to revise the corrections made in the manuscript. We have fixed all the issues raised in your comments.
Attached comments
